# Expression of miRNA-Targeted and Not-Targeted Reporter Genes Shows Mutual Influence and Intercellular Specificity

**DOI:** 10.3390/ijms232315059

**Published:** 2022-12-01

**Authors:** Dorota Hudy, Joanna Rzeszowska-Wolny

**Affiliations:** 1Department of Systems Biology and Engineering, Faculty of Automatics, Electronics, and Informatics, Silesian University of Technology, 44-100 Gliwice, Poland; 2Biotechnology Centre, Silesian University of Technology, 44-100 Gliwice, Poland

**Keywords:** regulation of translation, action of miRNAs, miR-21, miR-24, Let-7, anti-miRNAs

## Abstract

The regulation of translation by RNA-induced silencing complexes (RISCs) composed of Argonaute proteins and micro-RNAs is well established; however, the mechanisms underlying specific cellular responses to miRNAs and how specific complexes arise are not completely clear. To explore these questions, we performed experiments with *Renilla* and firefly luciferase reporter genes transfected in a psiCHECK-2 plasmid into human HCT116 or Me45 cells, where only the *Renilla* gene contained sequences targeted by microRNAs (miRNAs) in the 3′UTR. The effects of targeting were miRNA-specific; miRNA-21-5p caused strong inhibition of translation, whereas miRNA-24-3p or Let-7 family caused no change or an increase in reporter *Renilla* luciferase synthesis. The mRNA-protein complexes formed by transcripts regulated by different miRNAs differed from each other and were different in different cell types, as shown by sucrose gradient centrifugation. Unexpectedly, the presence of miRNA targets on *Renilla* transcripts also affected the expression of the co-transfected but non-targeted firefly luciferase gene in both cell types. *Renilla* and firefly transcripts were found in the same sucrose gradient fractions and specific anti-miRNA oligoribonucleotides, which influenced the expression of the *Renilla* gene, and also influenced that of firefly gene. These results suggest that, in addition to targeted transcripts, miRNAs may also modulate the expression of non-targeted transcripts, and using the latter to normalize the results may cause bias. We discuss some hypothetical mechanisms which could explain the observed miRNA-induced effects.

## 1. Introduction

The regulation of gene expression includes many mechanisms which modulate transcription and post-transcriptional processes, and is important to achieve and maintain the correct levels of proteins in cells. Much of the regulation takes place on messenger RNA (mRNA) during translation and post-transcriptionally, and the initiation step is critical for the regulation of translation [1,2,3,4]. A specific role in modulating the efficiency of translation of particular mRNAs is played by micro-RNAs (miRNAs), small 19–22 nucleotides-long RNAs that regulate the expression of most human mRNAs [5]. One miRNA can influence the translation of many mRNAs, and one mRNA can be affected by many miRNAs. MiRNAs interact with Argonaute proteins, of which there are four types in human cells (AGO1-4), to form the core of RNA-induced silencing complexes (RISCs) which recognize specific target mRNAs by complementary base pairing, normally in their 3′UTR. This causes downregulation of mRNA expression by inhibiting of translation or accelerating degradation of the mRNA. In human and animal cells, most miRNAs require only a “seed” region of perfect complementarity with the mRNA for translation inhibition [6]. In plant cells, miRNAs which are perfectly complementary to their mRNA target cause mRNA cleavage, which is rare in animal cells; in human cells only, AGO2 has an RNase domain [7].

The mechanism(s) by which miRNAs and RISC modulate translation is still not completely clear. The position of mismatches within or close to the seed sequence and cleavage site are determinant of the results of RISC action [8]. The RISC-induced degradation of mRNA depends on the type of Argonaute and the complementarity of a miRNA with its mRNA target. Depending on cell type, conditions, and miRNA–mRNA interactions, there are different pathways which lead to the inhibition of translation initiation and/or formation of mRNA-protein complexes visible as cytoplasmic foci, such as P-bodies or stress granules (SGs). P-bodies are large complexes containing mRNAs and proteins but devoid of translation machineries and enriched with translational repressors: factors involved in nonsense-mediated RNA decay, AU-rich element decay, miRNA silencing, and the de-capping protein Dcp1a. Stress granules have a different protein composition, containing the protein TIA1 but not Dcp1a [9]. It is not clear how miRNAs influence the creation of P-bodies or stress granules. All human AGO family members are found in P-bodies and SGs; the localization of AGO2 to P-bodies is not dependent on miRNA because it still appears in these structures when miRNA biogenesis is blocked. In contrast, AGO2 localization to SGs requires miRNAs [9].

Although the most common outcome of the action of miRNAs and RISC is the inhibition of gene expression, interactions which upregulate translation have been reported in the cases of some mRNAs for ribosomal proteins [10] and for activation of hepatitis C virus RNA [11,12]. Both effects were reported for transcripts with shortened poly(A), and depended on the growth conditions; interactions involving miRNA, AGO2, and FXR1 stimulated translation in quiescent cells but repressed translation in cycling cells [13,14,15].

Our study of the expression of miRNA-regulated reporter genes suggested that the expression of the unregulated reporter, used to normalize the results, also varies depending on the version of the gene targeted by the miRNA. Thus, in the present work, we studied another facet of regulation by miRNAs, the influence of miRNA-targeted sequences in the mRNA for a reporter gene on the expression of other genes. We compared the regulation of the expression of co-transfected miRNA-targeted and non-targeted reporter genes, and of the same reporter genes in two different cell types.

The results suggest that the presence of miRNA-targeted sequences in one gene can influence the expression of another non-targeted gene, and that different cell types respond to the same regulatory sequence in mRNA via the creation of different types of non-ribosomal complexes. We propose a model in which intermolecular interactions, including RNA–RNA interactions, may explain these effects.

## 2. Results

### 2.1. miR-21, miR-24 and Let-7 Influence the Expression Not Only of Targeted Renilla Reporter Genes but Also of Non-Targeted Firefly Luciferase Genes 

Using a common assay, we co-transfected HCT116 and Me45 cells with psiCHECK-2 plasmids containing two luciferase genes, a reference firefly gene and a reporter *Renilla* gene with or without eight tandem repeats of target sequences for the miRNAs Let-7, miR-21, or miR-24 in the 3′UTRs, and measured the levels of mRNAs and protein products (activity). 

MiRNA-21, -24, and Let-7 had relatively high expressions in HCT116 and Me45 cells; however, their concentrations were different. Microarray and RT-qPCR assays showed miRNA-21 to have the highest concentration in both cell lines, as seen in Figure 1.

Reporter genes were expressed very efficiently, and their levels exceeded the levels of housekeeping genes by about an order of magnitude. MiR-21-targeted sequences in the *Renilla* luciferase gene caused some decreases in the level of luciferase mRNA, a less significant decrease in Me45 cells, and a nearly complete inhibition of translation in both cell lines. The presence of an miR-24 target was accompanied by a decrease in mRNA level but a slight increase in protein level in Me45 cells, whereas, in HCT116 cells, both the reporter mRNA and protein levels decreased. A Let-7 target on the transcript resulted in a decreased level of mRNA and a decreased translation in Me45 cells; however, it did not significantly influence mRNA or increased protein levels in HCT116 cells (Figure 2). Conventionally, in similar transient transfection experiments, changes in expression of the firefly gene were used as indicators of transfection efficiency and the levels of its mRNA and protein served to normalize those of *Renilla* mRNA and protein. However, here, the levels of luciferase mRNA and protein (activity/cell) from the firefly gene, which was not targeted by any miRNA, also changed when cells were co-transfected with a *Renilla* luciferase gene regulated by a miRNA. In both HCT116 and Me45 cells, the changes in firefly mRNA were larger than those for transcripts targeted by miRNA-24 (Figure 2C,D, bars with lighter color). Firefly gene expression showed unexpected effects depending on the regulation of neighboring transcripts by miRNAs; for example, in Me45 cells, regulation of the *Renilla* gene by miR-24 or Let-7 was accompanied by a decrease in firefly luciferase mRNA together with an increase in its protein level (Figure 2C,D). The decrease in mRNA could be interpreted as the result of decreased transfection efficiency; however, the simultaneous increase in protein level in the same sample cannot be explained in this way and must result from a change in the regulation of translation of firefly mRNA.

In Table 1, we summarize the efficiencies of translation of the *Renilla* luciferase gene targeted by different miRNAs and of the co-transfected, non-targeted firefly luciferase genes, calculated as the quantity of protein (activity) per molecule of mRNA.

The translation efficiency of the *Renilla* transcript, which does not contain any regulatory sequences in its 3′UTR, is similar in both cell types; however, one has to remember that *Renilla* and firefly luciferases are different enzymes and that protein units based on luciferase activity are not comparable. The introduction of an miRNA-targeted sequence to the transcript changes the translation efficiency of both *Renilla* and firefly proteins. In HCT116 cells, the regulation of the *Renilla* transcripts by miRNA had less effect on the translation of firefly luciferase; however, this influence was more significant in Me45 cells. The presence of miR-21-targeted sequences on *Renilla* transcripts is correlated with a decreased translation efficiency of both *Renilla* and firefly transcripts in both cell types; however, the targeting of *Renilla* transcripts by miR-24 or Let-7 was correlated with a decrease in translation efficiency in HCT116 cells but an increase in Me45 cells, suggesting that the same reporter genes are regulated by different mechanisms in HCT116 and Me45 cells.

Changes in translation efficiency, calculated as the amount of protein activity per mRNA molecule, could result from an increase in the amount of protein translated from the same amount of mRNA (higher speed of translation, easier translation initiation with the same number of mRNA molecules), or, alternatively, from a decrease in the number of mRNA molecules with the same speed of translation and level of protein (higher degradation of non-translated mRNA).

### 2.2. Sucrose Gradient Centrifugation of Complexes Formed by Firefly and Renilla mRNAs

It was interesting to examine whether the different effects of sequences targeted by Let-7, miR-21, or miR-24 on luciferase mRNA and protein levels in the two cell lines were related to differences in the types of complexes containing mRNA during or after its translation. To separate mRNA-protein complexes of different sizes (ribosomes, polysomes, and non-complexed mRNA), cytoplasmic extracts were centrifuged in 15% to 45% sucrose gradients which were fractionated into 100 fractions, and the OD at 260 nm was measured in each fraction (Figure 3A). These small fractions were pooled into five larger fractions which contained enough material to perform assays of mRNA content; fraction 1 (top of the gradient) should contain RNA which is free or bound to proteins; fraction 2, small ribosomal subunits and complexes of similar size; fraction 3, large ribosomal subunits and monosomes; fraction 4, light polysomes and complexes of similar size; fraction 5, heavy polysomes and the heaviest non-ribosomal complexes (Figure 3A) [16]. The level of mRNA for *Renilla* and firefly luciferases in each fraction were assayed by RT-qPCR (Figure 3B,C).

Most of the *Renilla* luciferase mRNA was found in fraction 1 in both cell lines. The levels of reporter mRNA in the next fractions, 2, 3, 4, and 5, showed a gradual decrease, except in fraction 4, in HCT116 cells, which showed a slight increase in comparison to the preceding and following fractions. The high level of mRNA in the “free RNA” fraction was specific only for reporter gene mRNAs; transcripts for the non-reporter genes *GAPDH* and *RPL41* were distributed differently (Figure 4).

Next, we examined the distributions of miR-21-, miR-24-, and Let-7-targeted *Renilla* luciferase and untargeted firefly luciferase mRNAs in similar sucrose gradient fractions (Figure 5). The distributions of *Renilla* luciferase mRNA were different in each cell type and depended on the miRNA-targeted sequence. In HCT116 cells, more control and non-targeted *Renilla* reporter mRNA were found in fraction 4; however, when the reporter mRNA contained a Let-7 target, this increase was not seen (compare Figure 5A,D) and there was more reporter mRNA in fraction 1 (“free” mRNA or mRNA complexed with lower molecular weight proteins). These changes were accompanied by a slight increase in luciferase protein level and translation efficiency (Figure 2 and Table 2). In Me45 cells, the effect of a Let-7 target was different: the amount of “free” reporter mRNA decreased and it was more abundant in higher molecular weight fractions correlated with an increase in translation efficiency.

The presence of a miR-21 target in mRNA was correlated with less mRNA in fraction 1 and significantly more mRNA in heavy fractions in both cell lines; however, some differences between the cell lines were also obvious. In HCT116 cells, most of the miR-21-targeted mRNA sedimented in the heavier fraction 4, whereas, in Me45 cells, the peak was in fraction 2, containing complexes lighter than ribosomes. In the case of a miR-24-targeted sequence, the distribution of luciferase mRNA in polysomal fractions resembled that of control mRNA in both cell lines, except in Me45 cells, where more *Renilla* luciferase mRNA was present in fractions containing higher molecular weight complexes and less in fraction 1.

Firefly luciferase mRNA from each cell type also sedimented differently, and its distribution in the gradient changed according to the regulation of co-transfected *Renilla* transcripts. Complexes containing firefly mRNA from both cell types sedimented similarly to those formed by miRNA-targeted mRNAs in spite of the fact that each miRNA had a specific effect on the distribution of targeted *Renilla* mRNA. The above changes in the properties of complexes containing firefly luciferase mRNA were unexpected, and comparing the changes in firefly mRNA and protein levels with *Renilla* luciferase expression one could conclude that firefly luciferase expression was regulated by some mechanism depending on *Renilla* regulation, even in the absence of miRNA-targeted sequences.

### 2.3. Influence of Anti-miR-21, Anti-miR-24, and Different Anti-Let-7 Oligonucleotides on Expression of Reporter Luciferases

To confirm the impact of particular miRNAs on reporter gene expression, we introduced an anti-miRNA oligoribonucleotide for miRNAs which targeted the *Renilla* gene together with the reporter gene.

It is generally assumed that such anti-miR oligonucleotides inhibit miRNA action by forming hybrids with them and thereby prevent their interaction with mRNAs. The effect of anti-miR oligonucleotides for miRNA-21 and miRNA-24 on *Renilla* and firefly luciferase expression are shown in Figure 6, where expression changes are represented as the decimal logarithm of the ratio of mRNA or protein levels in the presence or absence of an anti-miR-oligonucleotide. As one could expect, the addition of an anti-miR-21 to the system with a miR-21-targeted *Renilla* luciferase caused an increase in the *Renilla* protein level in both cell types, while the untargeted firefly protein did not change significantly (Figure 6B,D). However, the response of mRNA levels was different in each cell type; in Me45 cells, the mRNA levels decreased, suggesting that some mRNA molecules are protected from degradation in these cells in the presence of active miR-21.

Anti-miRNA-24 oligonucleotides appeared to have an opposite effect on the protein and mRNA levels in HCT116 and Me45 cells (Figure 6E–H). Their presence also had a visible effect on the levels of the non-targeted firefly luciferase mRNA and protein.

The Let-7-targeted sequence in mRNA is particular in that it may interact with at least 12 different miRNAs belonging to the Let-7 group and encoded by different genes. HCT116 and Me45 cells differ in their levels of these different Let-7 miRNAs (Figure 7C,F). We used anti-miR oligonucleotides on Let-7 family members to see which were most effective in the regulation of reporter gene expression (Figure 7). The presence of an anti-miR oligonucleotide among different Let-7 family members had different effects in HCT116 and Me45 cells. In HCT116 cells, they caused a decrease in mRNA and protein levels, suggesting that, in these cells, interactions with any Let-7 have a protective effect on mRNA and that most such interactions may stimulate translation. In these cells, firefly luciferase expression also showed a similar response to different anti-Let-7 oligonucleotides (Figure 7A,B).

In Me45 cells, only the anti-Let-7a oligonucleotides had a similar inhibiting effect on *Renilla* mRNA, the other anti-Let-7s had an opposite effect on mRNA to that seen in HCT116 cells and, also, the influence on protein level was different. All Let-7 anti-miRs increased the level of *Renilla* luciferase in Me45 cells and (except for anti-Let-7d) decreased it in HCT116 cells. This result suggests differences in mechanisms regulating translation and mRNA stability between cell types and, also, different levels of participation of Let-7 family members in regulation. 

Anti-miR oligonucleotides had similar effects on the transcripts of non-targeted mRNAs and their translation in HCT116 cells, and caused a decrease in protein level, suggesting that interactions of *Renilla* transcripts with miRNAs somehow protect mRNA and also facilitate the translation of non-targeted mRNA. 

### 2.4. Proteins Potentially Engaged in Regulation of Translation Are Differently Expressed in Me45 and HCT116 Cells

Sucrose gradient centrifugation revealed that a particular mRNA targeted by the same miRNA was localized in different mRNA-protein complexes in Me45 and HCT116 cells (Figure 5). To explore if this could be related to differences in the expression of proteins participating in translation and its regulation, we compared the levels of transcripts for translation initiation factors and other potential regulatory proteins which may interact with AGO or initiation complexes by using Affymetrix microarrays and PCR methods (Figure 8 and Table 3).

The mRNAs for most initiation factors had similar levels in both cell types and, with some exceptions, the differences did not exceed two-fold. From those which had a higher expression in Me45 cells eukaryotic initiation factors (EIFs), EIF1, EIF2S2, some of EIF3, EIF4A1, EIF4EBP1, and EIF6 showed the largest difference. In HCT116 cells, a strikingly high expression was seen for EIF5A (Figure 8). It is difficult to hypothesize which of these differences could be directly responsible for the differences observed in our experiments, as most translation initiation factors are multifunctional RNA-binding proteins that participate not only in translation regulation but sometimes also in the regulation of other cellular processes.

The efficiency and regulation of translation also depends on the availability and activity of further proteins, which may constitute another layer which regulates the expression or activity of proteins directly participating in translation. In Table 2, we show examples of the mRNA levels for some of these proteins which were found in complexes regulating translation in both cell types.

Among these proteins, CNOT6, 7, and 8 are subunits of the CCR4-NOT core transcriptional and translational regulation complexes, and are important for the stabilization, cytoplasmic transport, and deadenylation of polyadenylated mRNAs. CNOT6 (CCR4a) and CNOT7 (CAF1) have RNase activity and bind to CNOT1, which is a platform that is able to bind further regulatory proteins [17]. ABCE1 is a highly conserved protein required for translation initiation and ribosome biogenesis, and is a ribonuclease L inhibitor; it also plays a role in translation termination and ribosome recycling by dissociating ribosomes into large and small subunits [18]. Another group of proteins present in Me45 and HCT116 cells that may be crucial for translation regulation is members of the large and highly abundant family of RNA-dependent DEAD-box helicases with ATPase activity (DDXs). Some members of this family participate in liquid–liquid phase separation and are present in RNA-containing phase-separated organelles in prokaryotes and eukaryotes [19]. Both HCT116 and Me45 cells show high levels of DDX5 helicase expression. Again, as for initiation factors, one cannot hypothesize which of the differently-expressed proteins may be the most important for intercellular differences in miRNA translation regulation; however, taken together, these data suggest that differences in the levels of proteins participating in regulating translation initiation could be responsible for the differences in reporter gene expression between HCT116 and Me45 cells.

## 3. Discussion

### 3.1. Specific miRNA Effects in Cells of the Same Type and Differences between the Effects of the Same miRNA in Different Cell Types

In this study, we compared the effects of different miRNAs and posed questions concerning the mechanisms underlying their specificity in the same cell type, and the differences between cell types. Reporter genes which differ only in their miRNA-targeted sequences, transfected into the same cell type containing the same sets of proteins which control translation, showed very different expression (Figure 2); the presence of a miRNA-21 target reduced translation strongly, whereas the presence of a miRNA-24 or a Let-7 target did not change or increase the synthesis of the reporter protein in a cell type-specific manner. In different cell types, the same miRNAs induce mRNA-protein complexes that have different sedimentation properties, differ in their influence on mRNA stability and translation efficiency, and react differently to the presence of anti-miR oligonucleotides (Figure 2, Figure 5, Figure 6 and Figure 7). 

These different effects could have many causes. The first may be a difference in the levels of the miRNAs themselves within the cell; in HCT116 cells, the miRNA-21 level was about 4 times higher than that of all Let-7s together, and 15 times higher than the level of miRNA-24. In Me45 cells, the differences between the levels of miR-21, miR-24, and total (summed) Let-7s were less pronounced (Figure 1). 

The higher level of miRNA-21 than of miRNA-24 in both cell types could explain the weaker inhibition of translation by miRNA-24 than miRNA-21, but not the stimulation of expression by miRNA-24 (Figure 2), which requires a different mechanism. The specificity of miRNA binding to various AGO proteins could result in the formation of different complexes and cause differences in the final effect of the miRNA; however, published studies exclude a specificity of different human AGOs [20,21].

Important factors influencing the operation of the RISC complex are interactions between miRNA and their targets in mRNA, such as the presence and position of mismatches which influence AGO’s action in translation repression and in endo- or exo-nucleolytic mRNA degradation [8,22]. Our reporter genes differed in the presence and distribution of mismatches; the miR-21 was perfectly complementary, whereas the miR-24 binding site was unpaired on nucleotide 13 and the 3′-most hexanucleotide, and the Let-7 binding site was unpaired on nucleotides 9–12 with every Let-7 family member, and these differences may play a role in eliciting a specific effect. However, not all miRNA-specific effects which were observed here can be directly explained by mismatches and their influence on RISC efficiency, because the effects of the same targets in different cell types are different (see Figure 2, where the influence of targets for miRNA-24 and Let-7 on protein level is opposite in HCT116 and Me45 cells). The intercellular differences in the effects of the same miRNAs are most probably caused by differences in the formation of RISC-complexes with other proteins, resulting from intercellular differences in protein concentrations. In human cells, complexes containing AGO proteins may contain a plethora of other proteins including TNRC6, PABC, HNRNP, HSP, IGFBP family proteins, helicases MOV10 and DEAD box-containing, cold shock domain-containing proteins, RBM14, TP53, translation elongation factors, and ribosomal proteins, as shown by immunoprecipitation [21,23,24,25]. In *C. elegans,* the same miRNA is found in RISC complexes of different compositions in different cell types [26]. RISC complexes with different compositions may co-exist in the same cell, and the frequencies of those which contain or do not contain helicases, nucleases, or nuclease inhibitors depend on the availability of these components at any given time. Further, proteins in AGO and TNRC6 complexes may be modified post-translationally by prolyl-hydroxylation, phosphorylation, ubiquitination, or poly-ADP-ribosylation, which may alter their interactions and influence miRNA activity at global or specific levels [27]. Different protein-RISC complexes may degrade mRNA or inhibit translation with or without mRNA degradation, or sometimes even stimulate translation depending on their composition (for example, through interactions of PABP-EIF4F). In our experiments, the presence of sequences targeted by Let-7 did not lead to mRNA degradation in HCT116 cells and was even associated with protection, because the addition of oligonucleotides that are complementary to various representatives of the Let-7 group caused a decrease in the level of the reporter gene mRNA (Figure 7A). 

The differences in function between individual miRNAs in the same cells may depend not only on the nucleotide sequence of the target but also on the conformation of the entire mRNA molecule, which always tends to fold and adopt spatial structures stabilized by complementary interactions. Such conformation of non-complexed “naked” *Renilla* luciferase transcripts containing targets for miRNA-21, miRNA-24, or Let-7 differed (not shown) as we analyzed them using the RNAStructure software [28,29]. However, one must remember that the transcript spatial conformations within the cell may significantly differ because of interaction with cellular proteins.

The differences between cell types in the effects of the same miRNA may be due to differences in the levels of this miRNA, but also to differences in the concentration and distribution of proteins that are capable of complexing with RISC, which is best seen in experiments with anti-miR oligonucleotides. Limiting the action of miRNA-21 by anti-miR in both cell types goes hand in hand with an increase in the level of protein, which is consistent with the concept of translation inhibition. However, in the case of mRNA, the effects in HCT116 and Me45 cells were opposite; in the former, the mRNA level increased while, in the latter, it decreased in the presence of the same anti-miR (Figure 6). This suggests that, despite similar effects on translation, the miRNA-21 initiated complexes have a different composition and interact differently with mRNA in each cell type. In the case of anti-miR-24 and different anti-Let-7 group members, the opposite effects in both cell types also concern protein levels (Figure 6 and Figure 7).

### 3.2. Pitfalls in Normalization of Results for miRNA-Targeted Gene Expression to Those for a Non-Targeted Gene

In transient transfection experiments, the presence of a target for miRNA on *Renilla* luciferase transcripts affected the expression efficiency of a non-targeted firefly luciferase gene. Conventionally, in studies using two co-transfected plasmids, the observed expression of a miRNA-targeted gene is “normalized” to the expression of a second gene regulated only by a promoter, in order to correct for differences in transfection efficiency. Here, we used a different strategy to analyze the results of studies on the influence of miRNAs on the expression of targeted reporter genes cloned in psiCHECK-2 plasmids; we analyzed, separately, the expression of both the co-transfected miRNA-regulated and the non-regulated gene, calculating and comparing the levels of mRNA and protein per single cell. This strategy revealed some unexpected features which are masked in the conventional normalization method; the insertion of a miRNA target into the *Renilla* luciferase gene caused changes not only in its own transcript and protein levels, but also in those of a co-transfected but non-targeted firefly reporter gene (Figure 2). Moreover, the targeted and non-targeted mRNAs are found in similarly sedimenting protein-mRNA complexes, as seen by sucrose gradient centrifugation, but differ depending on the type of miRNA target on the *Renilla* gene (Figure 5), and anti-miRNA oligonucleotides which theoretically should not influence the expression of mRNAs which are not regulated by these miRNAs also affect the expression of the firefly gene (Figure 6 and Figure 7).

Altogether, these results suggest that some amount of targeted and non-targeted *Renilla* and firefly mRNA molecules may participate in the same regulation process initiated by a miRNA. The occurrence of different types of mRNA in the same complexes is consistent with the observation of Morisaki et al. [30], who, using multi-epitope tags and antibody-based fluorescent probes to directly observe translation in single cells, showed that a certain amount of mRNA of different types may occur during translation in the same foci. We hypothesize that such co-regulation could result from the creation of compact complexes in a process similar to that which initiates stress granule or P-body formation by a combination of partially random RNA–RNA, protein–RNA, and protein–protein interactions initiated by the miRNA-directed binding of AGO [31,32,33,34,35,36,37]. The presence of AGO and trans RNA–RNA interactions appear to be necessary for creation of the core nucleation centers and, further, the P-bodies or stress granules [38,39,40]. In our experiments, the presence of miRNA targets and inhibition of translation are accompanied by the formation of polysome-sized complexes (Figure 5) which are significantly smaller than bodies visible by fluorescence microscopy. Nevertheless, the presence of miRNA-targeted and non-targeted mRNAs in these complexes would suggest similarity in the initiation of the P-body and translation inhibition processes, of which the latter could be further inhibited (for example by the presence of proteins such as NOT1, which inhibits the formation of P-bodies [41]). This hypothesis of partial randomness in creation of complexes by different RNAs and proteins in the presence of RISC could explain most of the effects observed in our experiments (including some dispersion of results between experiments); however, further experimental work is needed.

In some experiments, we observed an increased expression of non-targeted firefly mRNAs in the presence of targeted *Renilla* mRNAs (Figure 2D), which cannot be explained by the mechanism of compact complexes proposed above. We propose a further type of interaction which could potentially be influenced by RISC. The importance of mRNA circularization for efficient translation has been widely discussed (for example [2,42,43,44]), and is believed to be responsible for the more frequent re-initiation of circularized—compared to linear—mRNA [43,44,45]. The proximity of the 5′ and 3′ ends of an RNA molecule suggested by the structure of naked mRNA [42], if persisting inside the cell, would favor their interaction, and it has been proposed that the circularization of mRNA molecules can occur through the interaction of an EIF4F complex on the 5′ end with PABP proteins bound to the 3′ end, but is prevented by the binding of RISC [46,47,48]. On the other hand, in resting cells, there are also examples of stimulation of the translation of cellular and reporter genes by interactions of RISC containing AGO2 with miRNA and the FXR1 protein, which was explained by the formation of a complex joining the 5′ and 3′ ends of the mRNA, and which facilitates re-initiation of translation [14,49]. Such RNA circularization by protein complexes is usually assumed to occur only in *cis*, but *trans* 5′–3′ interactions between different mRNA molecules are not experimentally excluded. Following an earlier idea [46], we propose that the binding of RISC to the 3′ end of an mRNA is likely to create a hindrance to the circularization of single transcript but, in some types of complex, may facilitate *trans* 3′–5′ interactions of mRNA molecules, which could confer similar advantages for the activation of translation initiation as in circularized single molecules. This model of inter-molecular activation of translation would explain the lack of correlation between the amounts of EIF4F components and of PABP bound to the 5′ and 3′ ends of the same mRNA molecule found in some published experiments [50]; however, it needs to be proven by further experiments. 

Our results are consistent with a model where the translation of a population of specific mRNA molecules is regulated by the formation of several alternative and different types of mRNA-RISC complexes which contain single—as well as multiple—miRNA targeted and non-targeted mRNA molecules. At present, the model is mainly conceptual, and more detailed experimental studies of the relevant RNA–RNA and protein–RNA interactions are needed. New methods which allow for the exploration of the multiple conformations and interactions of RNAs in living cells and which follow processes ongoing on single molecules [51,52] create new possible directions for further investigations.

## 4. Materials and Methods

### 4.1. Cell Lines

HCT116 cells (human colorectal cancer, ATCC-CCL-247) and Me45 cells (human melanoma, Oncology Center, Gliwice described in [53]) were grown in standard conditions in DMEM/F12, (PAN Biotech, Aidenbach, Germany) supplemented with 10% fetal bovine serum (Eurx, Gdansk, Poland) and penicillin-streptomycin (Sigma-Aldrich, St. Louis, MO, USA) at 37 °C in a humidified atmosphere with 5% CO_2_, passaging and medium change were conducted every two days.

### 4.2. Plasmids

Cells were grown to 70–80% confluency before transfection (about two or three days). Cells were transfected with psiCHECK-2 plasmids (Promega, Madison, WI, USA) containing two luciferase genes, a reference firefly gene, and a reporter *Renilla* gene containing eight tandem repeats of target sequences for miRNAs Let-7 family, miR-21-5p, and miR-24-3p in their 3′UTRs. The Let-7 target sequence contained the motif TCGAGACTATACAAGGATCTACCTCAG with average complementarity of 71.75% to various mature let-7 family members, and the miR-21-5p target TCAACATCAGTCTGATAAGCTAAA was 100% complementary to the mature miR-21-5p sequence; the two last AAs form a spacer to limit complementarity for nonspecific binding. The miR-24-3p target sequence contained the motif ATACGACTGGTGAACTGAGCCG, 68% complementarity to the mature miR-24-3p. PsiCHECK-2 with Let-7 sequence targets was a kind gift from Martin Simard, Laval University, Quebec. Sequence synthesis, insertion, and verification were performed by BLIRT (Gdansk, Poland). The unmodified plasmid was used as an unregulated control.

### 4.3. Anti-microRNA Oligonucleotides (Anti-miRs)

Seven miRNA-inhibiting oligonucleotides (anti-miRs) were purchased from Integrated DNA Technology (Coralville, IA, USA). Their sequences were complementary to those in miRbase hsa-miR-21-5p: MIMAT0000076; hsa-miR-24-3p: MIMAT0000080; hsa-let-7a-5p: MIMAT0000062; hsa-let-7d-5p: MIMAT0000065; hsa-let-7f-5p: MIMAT0000067; hsa-let-7g-5p: MIMAT0000414; hsa-let-7i-5p: MIMAT0000415). Twenty pmol per well in 12-well plates were co-transfected with the appropriate plasmid.

### 4.4. Transfection Protocol

Cells were grown to 70–80% confluency before transfection (about two or three days). Transfection was performed with branched polyethylenimine (PEI) 1 mg/mL (Sigma-Aldrich, St. Louis, MO, USA). A total of 1 µg of plasmid DNA per well in 12-well plates or 8 µg for a T75 flask was mixed with 0.2 mL or with 1 mL serum-free medium, respectively, per well or flask. Next, 2.5 µL or 20 µL of PEI was added to a well or flask and the DNA-PEI solution was vortexed for 15 s and incubated at room temperature for 20 min, during which culture medium was exchanged for a serum-supplemented DMEM/F12 (1.8 mL per well, 9 mL per flask). The droplets of incubated DNA-PEI mix were added (0.2 mL serum-free medium + 1µg DNA + 2.5 µL PEI per well or 1 mL serum-free medium + 8 µg DNA + 20 µL PEI per flask) and, after 24 h of incubation in standard conditions, cells were harvested for protein or RNA assays. In experiments with sucrose gradient centrifugation, cells from flasks were transferred to 15 cm diameter dishes and incubated for the next 24 h (to avoid high confluency and inhibition of translation), after which the sucrose gradient centrifugation procedure was applied. In the experiments with anti-miR oligonucleotides, 20 pmol of anti-miR was added with corresponding plasmid DNA to the DNA-PEI mixture per well. 

### 4.5. Extraction and Assays of RNA 

Total and polysomal mRNAs were assayed by RT-qPCR method. RNA was extracted with a Total RNA Mini kit (A&A Biotechnology, Gdynia, Poland), and reverse transcription was done with a NG dART kit (Eurx, Gdansk, Poland) using both oligo(dT) and random hexamers according to the supplier’s protocol. qPCR was performed on a CFX96 Touch Real Time PCR System (Bio-Rad Laboratories Inc., Hercules, CA, USA) with the RT PCR Mix SYBR^®^ A kit (A&A Biotechnology, Gdynia, Poland) and primers from Table 3.

The numbers of reporter transcript molecules in cells were calculated using calibration curves constructed for each reporter gene on the basis of DNA PCR performed with known numbers of plasmid DNA molecules containing the analyzed gene.

The levels of miRNAs were measured with the RT-qPCR method. For miR-21-5p and Let-7a and f, reaction was done with TaqMan MicroRNA Reverse Transcription kit (Applied Biosystems, Foster City, CA, USA) and 5xRT/20xPCR TaqMan MicroRNA assay number 000397 for miR-21-5p, 000377 for Let-7a, and 000382 for Let-7f. The used references were U75 for miR21 (number 001219) and U6 for Let-7af (number 001093). All TaqMan reactions were done with TaqMan Universal PCR Master Mix (Applied Biosystems, Foster City, CA, USA). MiR-24-3p and Let-7i measurements were performed with miRNA 1st-Strand cDNA Synthesis kit (Agilent, Santa Clara, CA, USA) and with miRNA QPCR Master Mix (Agilent, Santa Clara, CA, USA) using universal reverse primer from miRNA 1st-Strand cDNA Synthesis kit and primers from Table 3 for miR-24-3p, Let-7i, and SNORD75 as reference.

### 4.6. Luciferase Assays

Luciferases were assayed by their activity in oxidizing luciferin with emission of luminescence using the Dual-Luciferase Reporter Assay System (Promega, Madison, WI, USA) according to the producer’s protocol. Luminescence was measured with an Infinite F200 Pro microplate reader (Tecan, Männdorf, Switzerland) on all-white flat-bottom 96-well plates (Corning Inc., Corning, NY, USA). Activities are expressed as arbitrary units (a.u.).

### 4.7. Sucrose Gradient Centrifugation

Three hours before harvesting cells, the culture medium was changed for fresh medium. Cells were incubated for 5 min with cycloheximide (Sigma-Aldrich, St. Louis, MO, USA cat. Num.:C1988-1G) (100 ng/mL), then washed with ice-cold PBS (PAN Biotech cat. Num.: P04-36500, Aidenbach, Germany) containing cycloheximide (100 ng/mL), detached by scraping, and immediately lysed or frozen in liquid nitrogen and stored for up to 6 weeks before lysis. Lysis buffer was polysome buffer (10 mM KCl, 2 mM MgCl_2_, 20 mM Tris-HCl pH 7.6, 1 mM DTT, 100 ng/mL cycloheximide) supplemented with 0.5% Triton X-100 and 2.5% glycerol, used ice cold. Extracts were centrifuged at 16,000× *g* for 10 min and the supernatant was layered on a pre-formed 15–45% sucrose gradient in polysome buffer in a 12 mL polyallomer tube. The gradient was centrifuged in a SW41ti rotor in an Optima XPN-100 Ultracentrifuge (Beckman Coulter Inc., Brea, CA, USA) at 4 °C for 4 h and 100 two-drop fractions were collected into Eppendorf tubes by piercing the tube bottom. The RNA content of fractions was measured on a NanoDrop 2000 (ThermoScientific, Waltham, MA, USA) at 260 nm, and on the basis of the absorbance profile, succeeding fractions were combined to create 5 larger fractions which should contain (from the top) free RNA, small ribosome subunits, monosomes with large ribosome subunits, light polysomes, and heavy polysomes, and these fractions served for further analyses.

### 4.8. Calculation of Cell Numbers Serving for Assays

The number of cells used in experiments was calculated from the total RNA in samples and the RNA content per cell, by measuring total RNA from 100,000 cells spectrophotometrically and counting cells by microscopy. The RNA/cell from triplicate assays was 20.97 ± 5.66 pg for HCT116 cells and 35.27 ± 5.78 pg for Me45 cells.

### 4.9. Microarray Analyses

To examine the influence of the levels of different eukaryotic initiation factors (EIFs) and proteins engaged in regulating translation, we compared the levels of their transcripts using culture conditions and Affymetrix microarrays described in our earlier studies [54,55]. The results are available in the ArrayExpress database under accession number E-MEXP-2623 [56]. miRNA levels were estimated by Agilent microarrays (G4870A SurePrint G3 Human v16 miRNA 8 × 60 k) and are available in the ArrayExpress database under accession number E-MTAB-5197 [56]. All data are MIAME compliant. Microarray data quality was assessed using the simpleaffy Bioconductor package [57]. Raw HG-U133A microarray data from two experiments on both cell lines were processed using Brainarray EntrezGene specific custom CDF (v22) [58] in R using the RMA algorithm implemented in the affy Bioconductor library [59]. Differentially expressed genes were identified using limma with a *q*-value correction for multiple testing [60,61]. Genes coding for proteins engaged in regulating translation was identified using GO terms [20,62]. To compare the levels of transcripts in HCT116 and Me45 cells, microarray results were normalized to the number of cells used for the assays.

### 4.10. Statistical Tests

Results are presented as mean ± SD from at least three separate assays performed on cells from the same passage. All calculations were performed using Microsoft Office 2016. Significances were determined by *t*-tests. In all experiments, significances were calculated in reference to reporter genes from an unregulated plasmid.

## Figures and Tables

**Figure 1 ijms-23-15059-f001:**
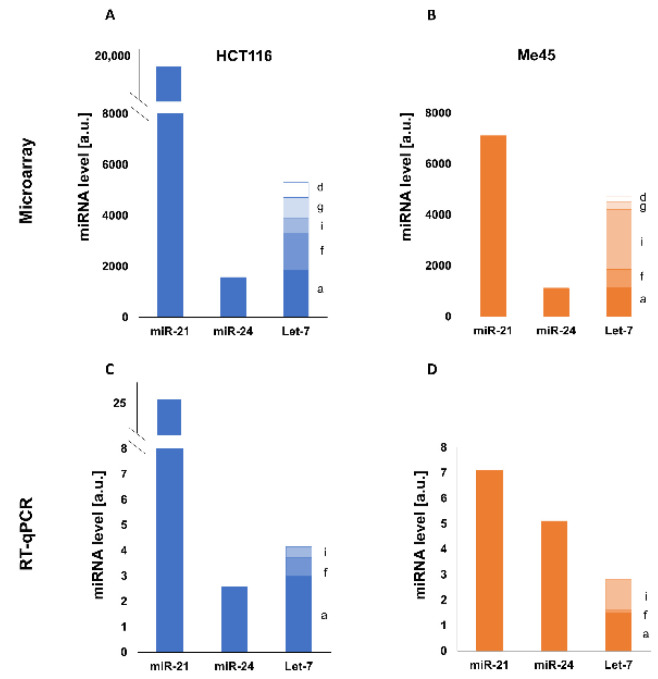
Levels of miRNA-21, miRNA-24, and Let-7 group miRNAs a, f, i, g, and d in HCT116 (**A**) and Me45 (**B**) cells measured by microarrays from E-MTAB-5197 data from ArrayExpress database and levels of miR-21, miR-24, and Let-7a,f, has in HCT116 (**C**) or Me45 (**D**) cells measured by RT-qPCR.

**Figure 2 ijms-23-15059-f002:**
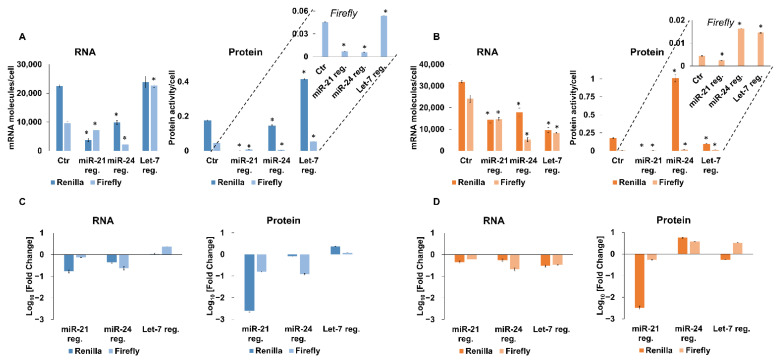
Expression of *Renilla* and firefly luciferase reporter genes transfected into HCT116 (**A**,**C**) or Me45 (**B**,**D**) cells. (**A**,**B**) show the levels of control and miRNA-targeted *Renilla*, and of non-targeted co-transfected firefly luciferase mRNA transcripts (molecules/cell) and proteins (activity/cell) in HCT116 and Me45 cells, respectively. (**C**,**D**) show fold changes in the levels of luciferase transcripts and proteins resulting from the presence of miRNA targets in the *Renilla* gene. The results are means from three independent experiments performed on the same cell populations and vertical bars show standard deviation. * indicate the significance with *p*-value < 0.05 compared to control transcripts.

**Figure 3 ijms-23-15059-f003:**
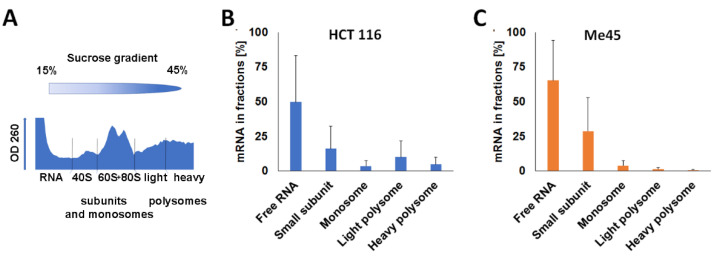
Distribution of mRNA for *Renilla* luciferase in fractions from sucrose gradients of cell extracts. (**A**) optical density profile of a gradient (HCT116 cells); (**B**,**C**) distribution of mRNA in sucrose gradient fractions of extracts from (**B**) HCT116 or (**C**) Me45 cells assessed by RT-qPCR (mean from three independent experiments, vertical lines show standard deviation).

**Figure 4 ijms-23-15059-f004:**
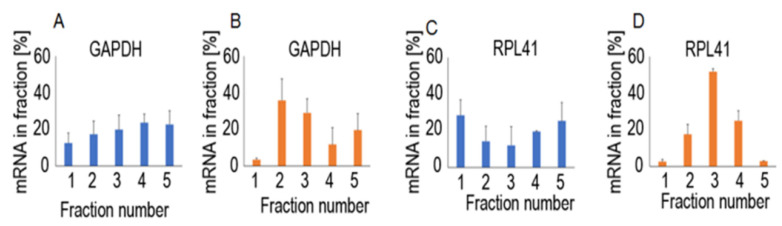
Distribution of mRNA for the *GAPDH* (**A**,**B**) and *RPL41* (**C**,**D**) genes in sucrose gradient fractions from HCT116 (**A**,**C**) and Me45 (**B**,**D**) cells transfected with psiCHECK-2 containing control reporter genes. Fractions number 1—free RNA, 2—small subunit, 3—monosome, 4—light polysome, 5—heavy polysome.

**Figure 5 ijms-23-15059-f005:**
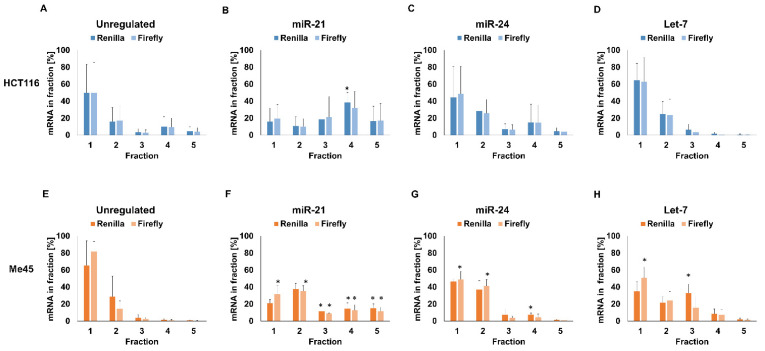
Distribution in sucrose gradients of control and miRNA-targeted *Renilla* and non-targeted firefly luciferase mRNAs from HCT116 or Me45 cells. (**A**,**E**): non-targeted *Renilla* mRNA, (**B**,**F**): *Renilla* mRNA targeted by miR-21, (**C**,**G**): *Renilla* mRNA targeted by miR-24, (**D**,**H**): *Renilla* mRNA targeted by Let-7group miRNAs. * indicate the significance with *p*-value < 0.05 compared to control transcripts (unregulated). Fractions number 1—free RNA, 2—small subunit, 3—monosome, 4—light polysome, 5—heavy polysome.

**Figure 6 ijms-23-15059-f006:**
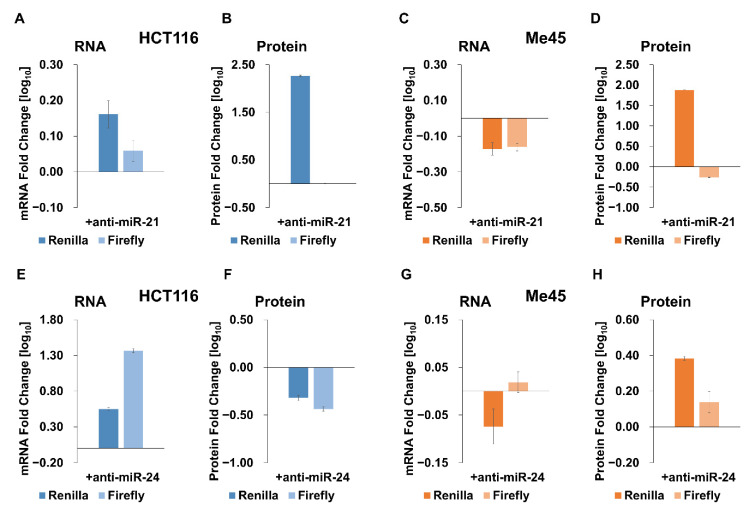
Influence of anti-miR oligonucleotides on miRNA-21 (**A**–**D**), or miRNA-24 (**E**–**H**) on the expression of miRNA-targeted *Renilla* and co-transfected non-targeted firefly reporter genes. Bar graphs represent the decimal logarithm of the ratio of mRNA (**A**,**C**,**E**,**G**) or protein (**B**,**D**,**F**,**H**) levels in the presence to these in absence of anti-miR oligonucleotides. Dark blue and dark orange columns show changes in *Renilla* luciferase expression, whereas light blue and light orange show values for firefly luciferase.

**Figure 7 ijms-23-15059-f007:**
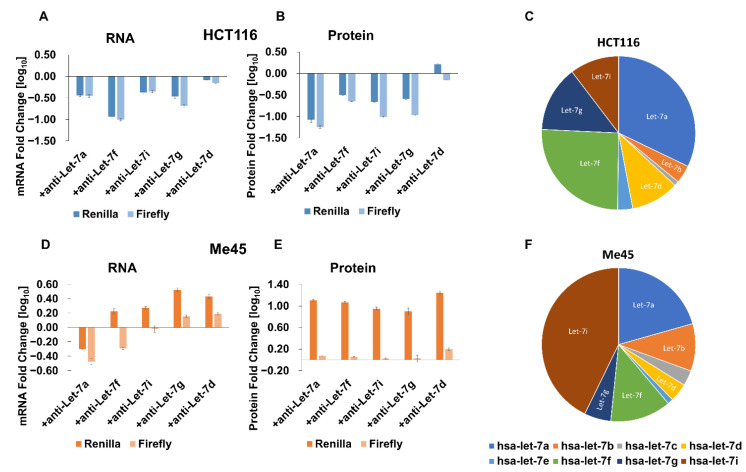
Influence of the inhibition of different Let-7 family members by anti-Let7 miRNA oligonucleotides on *Renilla* and firefly luciferase transcript (**A**,**D**), and protein (**B**,**E**) levels in HCT116 (**A**,**B**) and Me45 (**D**,**E**) cells. Bar graphs represent the decimal logarithm of the ratio of mRNA (**A**,**D**) or protein (**B**,**E**) levels in the presence of these in absence of an anti-miR oligonucleotides. The relative abundances of the different Let-7 family members in HCT116 and Me45 cells are presented as pie charts (**C**,**F**), respectively.

**Figure 8 ijms-23-15059-f008:**
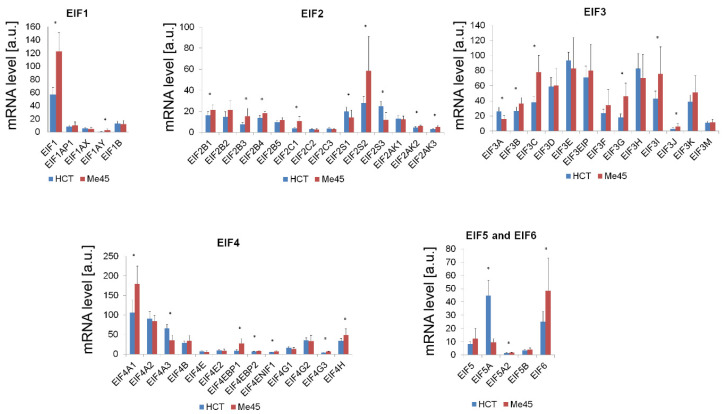
Levels of mRNAs for translation initiation factors cells assayed by microarrays (results available in ArrayExpress) normalized to cell numbers used for isolation of RNA. Each panel shows differences in a specific EIF group. The results are presented as mean values ± SD of three microarray experiments normalized to the number of cells, * denotes the statistical significance of the difference between HCT116 and Me45 cells (*p* < 0.05).

**Table 1 ijms-23-15059-t001:** Translation efficiency of miRNA-targeted *Renilla* or non-targeted co-transfected firefly transcripts [protein activity as a.u./mRNA molecule] *.

		No Target	miR-21	miR-24	Let-7
Me45 cells	*Renilla*	5.56 ± 0.10	0.04 ± 0.01	56.82 ± 5.62	10.15 ± 1.15
	firefly	0.18 ± 0.01	0.16 ± 0.01	3.23 ± 0.46	1.75 ± 0.06
HCT116 cells	*Renilla*	7.82 ± 0.21	0.12 ± 0.02	14.88 ± 0.94	17.45 ± 1.40
	firefly	4.71 ± 0.25	1.01 ± 0.05	2.46 ± 0.42	2.39 ± 0.05

* *Renilla* and firefly luciferases are different enzymes which use different substrates. The arbitrary units (a.u.) based on luminescence for each type are different and results for *Renilla* and firefly cannot be compared directly.

**Table 2 ijms-23-15059-t002:** Expression of genes coding for proteins (mRNA levels *) which may be engaged in modulation of translation.

Gene	HCT116 Cells	Me45 Cells
ABCE1	18.68 ± 3.05	11.79 ± 6.36
CNOT6/CCR4	5.66 ± 0.78	4.43 ± 2.35
CNOT7/CAF1	34.72 ± 6.09	17.75 ± 7.97
CNOT8/CAF2	6.19 ± 0.81	4.99 ± 1.84
CSDE1	19.50 ± 2.20	19.16 ± 4.97
DDX17	4.05 ± 1.74	4.49 ± 1.45
DDX3X	12.05 ± 1.73	10.60 ± 4.78
DDX3Y	0.85 ± 0.80	6.23 ± 2.39
DDX5	62.49 ± 15.75	84.26 ± 21.25
DICER1	1.67 ± 0.41	2.16 ± 0.38
EEF2	59.93 ± 10.37	104.99 ± 36.10
EIF2C1/AGO1	3.81 ± 0.74	10.81 ± 4.05
EIF2C2/AGO2	2.87 ± 0.52	2.71 ± 0.71
HNRNPH1	22.61 ± 5.34	21.80 ± 2.97
HNRNPL	20.70 ± 1.39	8.18 ± 2.19
HNRNPM	26.78 ± 5.77	20.08 ± 11.52
HNRNPU	32.20 ± 6.20	25.66 ± 7.63
HSP90AA1	149.67 ± 14.05	112.88 ± 20.58
HSP90AB1	186.57 ± 26.71	145.69 ± 33.39
HSPA1A	20.82 ± 10.19	10.42 ± 4.24
HSPA5	89.45 ± 23.53	114.94 ± 28.48
HSPA8	156.33 ± 28.72	149.77 ± 81.01
IGF2BP2	8.73 ± 0.93	17.40 ± 3.16
IGF2BP3	9.22 ± 1.37	8.03 ± 4.71
MKNK2	10.94 ± 2.08	7.51 ± 2.32
PABPC1	127.80 ± 22.45	144.75 ± 20.27
PABPC3	42.33 ± 7.50	31.14 ± 5.15
PABPC4	15.94 ± 3.45	51.53 ± 8.44
RBM14	6.77 ± 1.31	13.66 ± 3.41
SRP14	52.32 ± 12.92	78.13 ± 19.86
SYNCRIP	16.92 ± 2.65	8.68 ± 3.94
TNRC6B	4.38 ± 0.85	5.49 ± 0.89

* Assayed by Affymetrix microarrays; arbitrary units based on fluorescence.

**Table 3 ijms-23-15059-t003:** Real Time PCR primers.

Gene Name	Forward Primer	Reverse Primer
*Renilla* Luciferase	ACAAGTACCTCACCGCTTGG	GACACTCTCAGCATGGACGA
Firefly Luciferase	GCTAAGAGCACCCTGATCG	CCTCTGGGGTAATCAGAATGG
GAPDH	TTTGGCTACAGCAACAGGGTG	TTCCTCTTGTGCTCTTGCTGG
RPL41	TCCTGCGTTGGGATTCCGTG	ACGGTGCAACAAGCTAGCGG
hsa-Let-7i-5p	GTGAGGTAGTAGTTTGTGCTGTT	Universal from miRNA 1st-Strand cDNA Synthesis kit
hsa-miR-24-3p	TGGCTCAGTTCAGCAGGAACA	Universal from miRNA 1st-Strand cDNA Synthesis kit
U75	AGCCTGTGATGCTTTAAGAGTAG	Universal from miRNA 1st-Strand cDNA Synthesis kit

## Data Availability

The main data supporting the results of this article are included within the article, and detailed explanations can be provided on request. Microarray data are available in the ArrayExpress database, accession numbers: E-MEXP-2623, E-MTAB-5197.

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
