# Peer review of "Expression of miRNA-Targeted and Not-Targeted Reporter Genes Shows Mutual Influence and Intercellular Specificity"

_ijms, 2022, doi:10.3390/ijms232315059_

Round 1

Reviewer 1 Report

This study is well conducted, organised and presented, Thus can be accepted in the current form. However, a brief spelling check is needed. I believe the authors can take care of this in the final proofs.

Author Response

Thank you for the positive opinion. We checked the spelling (not only) and we hope there are no more mistakes. 

Reviewer 2 Report

In the manuscript by Hudy and Rzeszowska-Wolny, the authors test the effects of miRNA mediated downregulated on targeted and non-targeted reporter constructs by using different cell lines and find that there are off-target effects which were miRNA specific. The effects on mRNA and protein levels were dependent on the cell line used. Presence of free mRNA or RNA bound to proteins in different complexes was assayed by using sucrose gradient centrifugation. Both the firefly and the renilla luciferase mRNAs were found to be in same complexes in a cell type specific fashion. The findings presented here thus serve as a caution for experiments which utilize the transfection of non-targeted luciferase as a control. Most of the experiments in this paper are well conducted even though there is a lot of speculation about mechanistic explanation in the discussion without necessary experiments to validate the speculations and hence the reviewer recommends the authors to tone down the discussion. My comments regarding this paper are summarized below.

I wonder whether the presence of such high amounts of free mRNA and the off target down regulation which is observed is a result of overexpression. In that regards did the authors check in Fig1 what the levels of GAPDH or RPL41 or other housekeeping highly expressing genes were. It would be good to include that data in the paper to see if the off-target effects were specific to the firefly luciferase transcript.

For the experiments involving sucrose gradient centrifugation of mRNA-protein complexes the authors should indicate the reference which was followed otherwise must show some evidence to support the different complexes in their fractionation which they claim should be there in individual fractions. Moreover, it appears that the sucrose gradient fractionation of endogenous transcripts of GAPDH and RPL41 in Fig3 was done from control cells not transfected with the luciferase reporter and miRNAs. It would be useful to know the distribution of these mRNAs in mRNA complexes in conditions similar to those presented in Fig4.

The paper will benefit from a bit more detailing in the results section and the figure legends. Please explain that why the OD260 measurements for the fractions measuring the nucleic acids in Fig2A does not match with the mRNA content in Fig2B.For example the peak for monosome at OD260 looks stronger than 40S subunit but the mRNA content plotted in Fig2B is the other way round.

The authors must elaborate the Figure legends in bit more details so that it is easy to follow what is being represented in the figures. For example, there is no description for Fig3. There is no harm in describing what the sucrose gradient fractions are again in Fig3. Similarly, for most of the other figures use the figure legends to explain the results better.

Expand EIF when used for the first time.

Reviewer 3 Report

In this study the regulatory mechanism of miR-24, let-7 and miR-21 was studied using a luciferase expression system. The topic has great relevance because growing interest has been invested to miRNAs and their regulatory role is still not completely understood. The manuscript raises an interesting phenomenon. However, it contains several limitations.

My major comments:

1)      The authors applied 2 different cell lines (HCT116 and Me45) that differed in the expression of miRNAs including miR-21, miR-24 and let-7 (as presented in Figure 8). Furthermore, their relative expression was also different. Because the conclusions relied on endogenous miRNA expression only - and no miRNA mimic system was applied in order to control miRNA content - the significance of endogenous miRNA expression of the cell lines has great importance and might be responsible for most of the observed changes. For this reason the endogenous miRNA expression of the cell lines needs to be clearly presented in the beginning of the study and all the conclusions need to be reconsidered in the light of these values. Furthermore, it is not clear what microarray experiment was used to determine the miRNA levels that are presented in Figure 8. Note that qPCR is considered to be the most reliable method for the quantification of single miRNAs.

2)      It is not presented what form of miR-21, miR-24 and let-7 was studied (3p or 5p version?). Furthermore, miR-24 and let-7 have several versions.

3)      In the manuscript it is concluded that the expression of targeted and non-targeted reporter genes were also decreased after transfection. What was the negative control that was applied during transfection? Were the results compared to this? It has great significance because the stress induced by the transfection procedure might be resulted in the decrease of mRNA/protein production.

4)      It is required to clearly present and reconsider the normalization procedure because this might be responsible for most of the confusing results.

5)      It is not clear what the significance of experiments based on sucrose gradient distribution is.

6)      The discussion is full of speculations. No data are presented on RISC associated proteins or mRNA circularization.

Minor comments:

7)      Significance levels need to presented on the Figures.

8)      What values are presented in Table 2?

9)      The protein expression of Figure 1 A and B seems to be strange in this form.

10)   More details are required in the Materials and methods section e.g. culturing conditions in the case of the cell lines, transfection procedure, microarray analysis (what samples were used for this?).

Reviewer 4 Report

Authors study the effect of three different mi-RNA targeting the Renilla gene on the Firefly luciferase reporter gene in two different cells lines. Interestingly, although the effects observed are quite heterogeneous  depending on either the mi-RNA and cell line used, they observe that the expression of the luciferase is affected, indicating that the traditional normalization methods should be revised. Results are nicely presented with high quality figures.

Even though the discussion section is quite speculative and more experiments should be performed, I recommend its publication.

Reviewer 5 Report

In this MS, authors suggest that miRNA-21 caused strong inhibition of translation whereas miRNA-24 or Let-7 caused no change or an increase in reporter Renilla luciferase synthesis and also miRNA targets on Renilla transcripts also affected expression of the co-transfected but non-targeted Firefly luciferase gene in both cell types, indicating miRNAs may also modulate the expression of non-targeted transcripts. Despite interesting data, it has some concerns as follows:

1.     Rewrite the rationale of this study compared to previous evidences to show the significance and novelty of your work.

2.     Check English expression and careless mistakes e.g. Total and polysomal mRNA were assayed by RT-qPCR. The number of cells used in experiments were calculated

3.     Validate the above results by Western blotting or RT-PCR, if possible, since your results are well expected or reported  

Round 2

Reviewer 3 Report

In contrast to the fact that the authors made some minor modifications in the previous version of the manuscript most of my major concerns were not answered properly. I do still think that the following drawbacks need revision in the manuscript:

-        The expression of endogenous miRNAs has exceptional significance that needs to be considered more in the conclusion section. Furthermore, it is necessary to demonstrate the characteristics (e.g. miRNA expressions) of applied cell lines in the study before presenting the other experiments. Another option could be transfecting the cells with miRNA mimics in different doses and determining their dose-dependent effect to the expression of targeted and non-targeted reporter genes.

-        A more precise description is required for the transfection procedure and for data normalization to the materials and methods section.

-        More explanation is required for the significance of results based on sucrose-gradient experiments in both the results and discussion sections.

-        The conclusions need to rely on the results and not on speculations.

Reviewer 5 Report

They responded well to my comments

Author Response

Thank you

Round 3

Reviewer 3 Report

The authors relected to my major concerns.